# Latent Merging: Dynamic and Reversible Composition of Large Language Models

## Abstract

Weight merging is a common way to combine large language models, but its static and irreversible nature limits controllability and can destabilize behavior. We propose Latent Merging, which composes models in the hidden-representation space to enable dynamic, reversible, and layer-wise control without modifying weights. We unify classic operators —linear/spherical interpolation, and regularized means—under a single operator view and extend them from parameters to latents. We derive local second-order bounds on loss change that account for RMSNorm nonlinearity and head mismatch, yielding practical guidance (merge later; align heads) and stability guarantees. In data-free evaluation on Qwen2.5-7B-Instruct and its fine-tuned derivative, Latent Merging consistently surpasses weight merging on JudgeBench across reasoning, knowledge, mathematics, and coding; for example, SLERP attains 74.8 overall vs. 25.3 for weight merging. Representation analyses show stronger semantic preservation (cosine/CKA $> 0.8$), and layer-wise studies indicate that higher mixing ratios ($\alpha \approx 0.75$) in deeper layers work best while remaining bounded by source-model capacity. Latent Merging reframes model composition as controlling states rather than rewriting weights, offering a practical, theory-grounded path to controllable and interpretable LLM integration.

## 1 Introduction

Large language models (LLMs) are continuously expanding to adapt to diverse domains and tasks, and within this process, methodologies for integrating the capabilities of different models have been actively investigated. A representative approach is **weight merging**, where techniques such as model soup (Wortsman et al., 2022a), and Fisher-weighted merging (Matena & Raffel, 2022) synthesize knowledge from multiple models in parameter space to construct a new model. These methods are noteworthy for their simplicity and effectiveness, contributing to performance improvements and enabling multi-domain adaptability.

Nevertheless, weight merging intrinsically involves **irreversible and global modifications** (Wortsman et al., 2022a; Matena & Raffel, 2022). Once parameters are merged, restoration is not straightforward, and additional checkpoints must be generated and distributed. Flexible control conditioned on inputs or contexts is also limited. Furthermore, selectively combining only particular capabilities of a model or dynamically adapting the merging behavior according to task requirements remains challenging. More fundamentally, weight merging treats models as *static artifacts* fixed parameter sets that must be rewritten to alter behavior. As a result, merging is restricted to an *offline operation*, yielding a single static outcome.

To address these limitations, we turn our attention to the **latent space**, i.e., the hidden representations of models. In Transformer architectures, hidden states serve as the dynamic substrate through which inputs are interpreted and outputs are generated (Vaswani et al., 2017). Manipulating this space enables direct control of model behavior at inference time. In particular, merging or interpolating hidden states reformulates composition from *modifying functions* to *controlling ongoing computations*, thereby allowing **synthesis of model properties without altering parameters**. This naturally extends the notion of weight merging into the latent domain, which we term *latent merging*.

In this paper, we propose a latent merging framework that generalizes weight-space merging techniques to hidden representations. Specifically, operations such as linear averaging and orthogonalization-based interpolation can be directly applied in latent space and executed dynamically. The proposed framework enables **reversibility**, **conditional control**, and **localized intervention** without modifying model parameters, thus establishing a new paradigm for model composition.

The contributions of this work are summarized as follows.

- **Conceptual Contribution**: We introduce a novel perspective that establishes a correspondence between weight merging and latent merging.

- **Framework-level Contribution**: We establish a unified latent merging framework that enables dynamic, reversible, and localized control of model behavior in hidden representation space, going beyond the static and irreversible nature of weight merging.

- **Empirical Contribution**: Through experiments on domain transfer and style control tasks, we demonstrate that latent merging improves the performance stability trade-off compared to weight merging.

In summary, latent merging shifts the focus from modifying the model to controlling its states. Unlike the global and irreversible nature of weight merging, latent merging enables fine-grained and reversible manipulation within the hidden representation space, opening a new paradigm for model composition in LLMs. Importantly, even by directly applying conventional weight merging techniques in latent space, strong performance improvements can already be achieved.

## 2 RELATED WORK

**Weight Merging.** A line of research has explored directly merging models in weight space. Wortsman et al. (2022b) introduced *Model Soup*, which averages the weights of multiple fine-tuned models derived from the same base model, improving accuracy and out-of-distribution robustness. However, this method becomes unstable when models are less similar, and the resulting merged model is irreversible.

Matena & Raffel (2022) employed Fisher information to weight parameters by their importance, improving stability, while Jin et al. (2023) cast merging as a regression problem to obtain more robust solutions. Nonetheless, these approaches still result in a single fixed model, lacking dynamic control at inference.

**Latent Space Manipulation.** Another direction manipulates latent representations instead of weights. Dathathri et al. (2020) introduced *Plug and Play LMs*, which adjust hidden states guided by an external classifier to enforce desired attributes such as sentiment. Subramani et al. (2022) extracted latent steering vectors that, when added to hidden states, control sentence style or semantics. Kumar et al. (2023) trained lightweight transformation modules using contrastive learning, enabling controllable modifications (e.g., tone, toxicity, reading level) that are composable and reversible. In vision, GAN studies have shown that interpolating or shifting latent codes leads to semantically meaningful changes. Together, these works suggest that hidden spaces provide controllability and compositionality that weight merging lacks.

**Modular and Compositional LLMs.** A third research strand focuses on modular architectures that preserve task-specific abilities. Houlsby et al. (2019) and Pfeiffer et al. (2021) introduced adapters and *AdapterFusion*, where task-specific adapters are combined through fusion layers to enable non-destructive task composition. Shazeer et al. (2017); Fedus et al. (2022) explored Mixture-of-Experts (MoE) architectures, where a gating function routes tokens to experts, achieving scalability and conditional control. Prompt-based approaches such as Lester et al. (2021) and Asai et al. (2022) learn soft prompts per task and combine them using attention, enabling multi-task transfer with reversible control. More recently, Wu et al. (2024) proposed *Mixture of LoRA Experts* (*MoLE*), which organizes LoRA modules into experts and employs gating to achieve conditional and compositional control while avoiding instability from naive merging. These approaches collectively provide reversibility, conditional control, and stability, addressing structural limitations of weight merging.

**Relation to Our Work.** Weight merging is simple but irreversible and uncontrollable; latent manipulation and modular methods provide controllability but mainly focus on attributes or modular training. Our proposed *Latent Merging* differs by applying generalized merging operators directly in representation space, combining the simplicity of weight merging with the controllability of latent and modular approaches.

## 3 METHODOLOGY

Our approach is built upon the observation that existing weight merging techniques can be understood as special cases of a single, more general composition framework. Rather than treating linear interpolation or spherical interpolation as independent heuristics, we cast them under a unified operator formalism. This abstraction not only clarifies the relationships among prior methods, but also provides a natural pathway to extend merging from parameter space to latent representation space, thereby enabling new

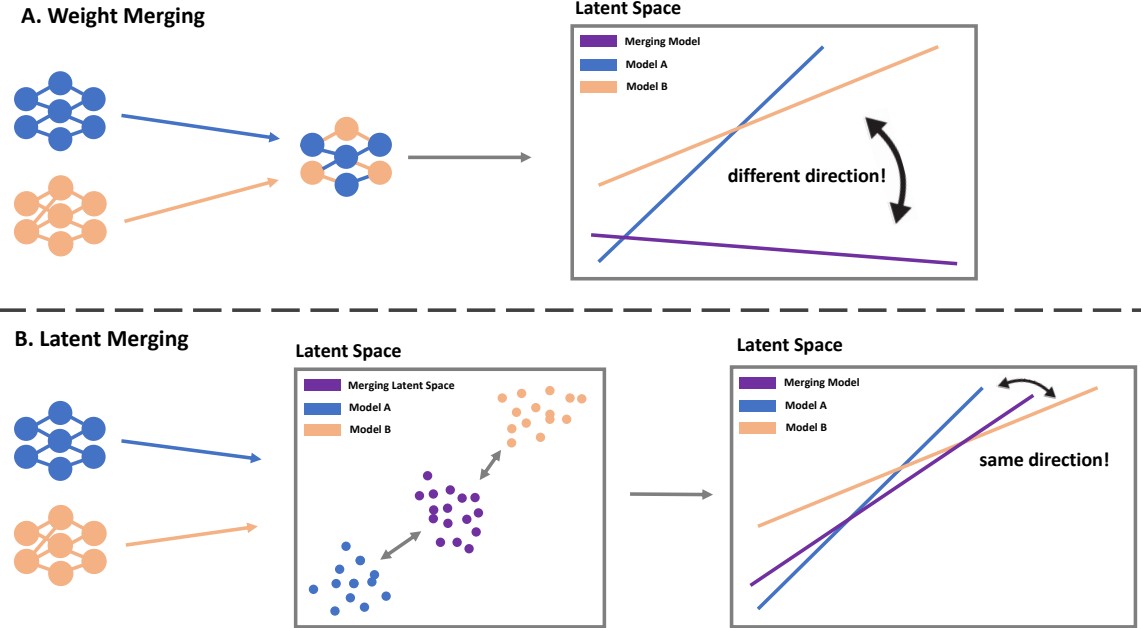

Figure 1: Overview of the method. (A) Illustration of weight merging, where parameters from multiple models are composed under different operators. (B) Illustration of latent-space merging, where representations are dynamically combined to enable reversible model composition. Together, the two panels highlight the distinct but complementary roles of parameter- and latent-level composition.

forms of dynamic and reversible model composition. An overview of the experimental setup is shown in Figure 1.

### 3.1 GENERALIZED MERGING OPERATOR

We begin by formalizing weight merging as a generalized operator that subsumes a wide range of existing techniques. Given two models $A$ and $B$ with parameters $\theta_A$ and $\theta_B$, a merging operation can be expressed as

$$\theta' = \mathcal{M}(\theta_A, \theta_B; \alpha, \mathcal{G}), \tag{1}$$

where $\alpha \in [0, 1]$ denotes the mixing coefficient, $\mathcal{G}$ specifies the geometry in which the merge is performed and $\mathcal{M}$ represents the merging operator defined with respect to $\mathcal{G}$.

Classical approaches such as linear interpolation (LERP), spherical interpolation (SLERP), and Fisher-weighted averaging can all be recovered as particular instantiations of this operator. Viewed this way, weight merging is not a collection of ad-hoc procedures, but rather an instantiation of a general family of parameter-space composition rules.

### 3.2 LATENT MERGING FRAMEWORK

We extend the notion of merging from parameters to hidden representations. Let $h_A$ and $h_B$ denote the latent states produced by two models (or training trajectories) at a given layer or time step. We define latent merging as

$$h' = \mathcal{M}(h_A, h_B; \alpha, \mathcal{G}), \tag{2}$$

where $\mathcal{M}$ denotes a merging operator (e.g., LERP, SLERP, RegMean) and $\alpha \in [0, 1]$ controls the mixing ratio. This formulation directly composes internal representations rather than model parameters, thereby preserving the expressive operator family while avoiding the rigidity of weight-level modifications.

A crucial factor lies in how merged representations are mapped to outputs. Modern LLMs apply a nonlinear pre-normalization $r(h)$ (e.g., RMSNorm) prior to the final linear projection, yielding logits of the form $Wr(h) + b$. This architectural detail alters the theoretical landscape: the loss bound is no longer a direct Jensen-type interpolation but instead includes correction terms that reflect the curvature of the normalization and potential mismatches across LM heads. These considerations help explain why merging at later layers tends to be more robust and why head alignment often plays a critical role.

### 3.3 Theoretical Guarantees under Nonlinear Pre-heads

We study latent merging in the presence of a nonlinear pre-head $r : \mathbb{R}^d \to \mathbb{R}^d$ (e.g., RMSNorm) followed by a linear LM head. For model $A$, let $g_A(h) = W_A\, r(h) + b_A$, and define $z_A^{(A)} = g_A(h_A)$ and $z_B^{(A)} = g_A(h_B)$. Given two hidden states $h_A, h_B$ at a fixed layer, we merge

$$h'_\alpha = (1-\alpha)h_A + \alpha h_B, \qquad \alpha \in [0,1],$$

and evaluate with a convex, $\beta$-smooth logit-based loss $\ell$ (e.g., cross-entropy). Optionally assume $\|\nabla \ell(u)\| \le G$ along the segment $u = (1-\alpha)z_A^{(A)} + \alpha z_B^{(A)}$.

**Assumption (local smoothness).** $r$ is $\mathcal{C}^2$ on a neighborhood of the segment $[h_A, h_B]$ and its Jacobian is $K_r$-Lipschitz there. Equivalently, for $g_A(h) = W_A r(h) + b_A$, the Jacobian of $g_A$ is $K_g$-Lipschitz with $K_g \le \|W_A\|_{\mathrm{op}} K_r$.

**Theorem 1** (Local second-order bound). *Under the assumptions above, for $h'_\alpha = (1-\alpha)h_A + \alpha h_B$,*

$$\begin{aligned}
\ell\big(g_A(h'_\alpha)\big) &\le (1-\alpha)\,\ell\big(z_A^{(A)}\big) + \alpha\,\ell\big(z_B^{(A)}\big) \\
&\quad + \frac{GK_g}{2}\,\alpha(1-\alpha)\,\|h_B - h_A\|^2 \;+\; \frac{\beta K_g^2}{8}\,\alpha^2(1-\alpha)^2\,\|h_B - h_A\|^4.
\end{aligned}$$

**Comparator vs. native heads (shared $r$).** If model $B$ uses the same $r$ with head $g_B(h) = W_B r(h) + b_B$, define the head-mismatch at $h_B$:

$$\Delta_{\mathrm{head}}(B \to A) = \big\|(W_A - W_B)\,r(h_B) + (b_A - b_B)\big\|.$$

Then

$$\begin{aligned}
\ell\big(g_A(h'_\alpha)\big) &\le (1-\alpha)\,\ell\big(g_A(h_A)\big) + \alpha\,\ell\big(g_B(h_B)\big) \\
&\quad + \underbrace{\tfrac{GK_g}{2}\,\alpha(1-\alpha)\,\|h_B - h_A\|^2 \;+\; \tfrac{\beta K_g^2}{8}\,\alpha^2(1-\alpha)^2\,\|h_B - h_A\|^4}_{\text{pre-head curvature}} \qquad\qquad (3) \\
&\quad + \underbrace{\alpha\, G\, \Delta_{\mathrm{head}}(B \to A) \;+\; \alpha\,\tfrac{\beta}{2}\,\Delta_{\mathrm{head}}(B \to A)^2}_{\text{head mismatch}}.
\end{aligned}$$

**Corollary (linear limit).** If $r$ is identity, then $K_g = 0$, the curvature terms in equation 3 vanish, and we recover the exact Jensen guarantee for linear heads.

**Practical guidance.** The correction scales as $O\big(\alpha(1-\alpha)\|h_B - h_A\|^2\big)$ plus a quartic term. Thus merging later, keeping $\|h_B - h_A\|$ small via normalization or SLERP, and aligning LM heads $(W, b)$ improves robustness. Full derivations and extensions (including SLERP small-angle analysis, and RegMean via proximal maps) appear in the Appendix A.

## 4 Experimental Setup

In this study, we adopt a setting that does not rely on any additional data, such as pre-training or post-training. This choice is motivated by the fact that introducing new datasets for further training makes it difficult to disentangle whether performance improvements come from the data itself or from the merging technique. by establishing a data-independent environment, we are able to isolate and measure the intrinsic effect of merging methods, providing the fairest conditions for comparing weight merging and latent merging.

Accordingly, our experiments focus exclusively on merging techniques without any auxiliary data, with the goal of directly examining how their structural differences influence performance.

The experimental evaluation is divided into three parts, each designed to investigate different aspects of the proposed method.

### A. Comparative Evaluation of Weight Merging and Latent Merging

A critical aspect of evaluating merging techniques lies in the choice of benchmark. Conventional accuracy-based evaluations, most notably MMLU-Pro Wang et al. (2024), impose strict formatting requirements that can cause scores to fluctuate dramatically depending on surface-level output properties. Such metrics are particularly sensitive to instabilities introduced by weight merging, including repetitive or

malformed generations, which in turn obscure the model's underlying capabilities. As a result, relying solely on exact-match accuracy provides an incomplete and potentially misleading comparison between weight merging and latent merging.

To mitigate these limitations, we adopt the **JudgeBench** Tan et al. (2024) framework. JudgeBench employs a large language model as a reference judge to conduct pairwise comparisons of system outputs. This evaluation paradigm extends beyond correctness alone, incorporating dimensions such as *fluency*, *coherence*, and *semantic adequacy*. By capturing these multifaceted aspects of generation quality, JudgeBench provides a more faithful and robust assessment of merged models.

In practice, JudgeBench spans four representative domains: *Knowledge* (drawn from MMLU-Pro Wang et al. (2024)), *Reasoning* and *Math* (sourced from LiveBench White et al. (2024)), and *Coding* (from LiveCodeBench Jain et al. (2024)). Together these tasks ensure that evaluation covers factual recall, multi-step reasoning, mathematical problem solving, and code generation. Accordingly, all comparative experiments in this study are conducted under JudgeBench, ensuring that observed differences reflect the intrinsic structural properties of weight merging and latent merging rather than artifacts induced by brittle accuracy metrics.

## B. Latent Space Similarity Analysis

The second experiment evaluates whether latent merging improves not only accuracy-level performance but also the stability and preservation of internal representation spaces. A successful merging process should produce latent representations that retain the semantic distributions of the original models while also reflecting the characteristics of both models in a balanced manner.

**Setup & metrics.** Given hidden states $\mathbf{h}_A, \mathbf{h}_B$ from two source models and a merged state $\mathbf{h}_M$ at the same layer and token, we compare latent-space vs. weight-space merging using three complementary metrics:

- **Midness.** With cosine distance $d(\mathbf{x}, \mathbf{y}) = 1 - \cos(\mathbf{x}, \mathbf{y})$, we define

$$\text{Midness}(\mathbf{h}_A, \mathbf{h}_B, \mathbf{h}_M) = 1 - \frac{|d(\mathbf{h}_M, \mathbf{h}_A) - d(\mathbf{h}_M, \mathbf{h}_B)|}{d(\mathbf{h}_A, \mathbf{h}_B) + \varepsilon},$$

  which measures how balanced $\mathbf{h}_M$ lies between $\mathbf{h}_A$ and $\mathbf{h}_B$ (higher is better).

- **Arc Ratio.** On unit-normalized states, let $\theta(\mathbf{x}, \mathbf{y}) = \arccos\big(\cos(\mathbf{x}, \mathbf{y})\big)$. The ratio

$$\text{ArcRatio} = \frac{\theta(\mathbf{h}_A, \mathbf{h}_M)}{\theta(\mathbf{h}_A, \mathbf{h}_B)}$$

  is ideally 0.5 if $\mathbf{h}_M$ lies on the geodesic midpoint; we report both the raw value and its deviation $|\text{ArcRatio} - 0.5|$ (lower is better).

- **Centered Kernel Alignment (CKA) Kornblith et al. (2019).** We compute linear CKA layer-wise between representation matrices of the merged and source models and average across layers/tokens. Unlike simple pointwise similarity, CKA captures structural alignment across the full representation space, making it a robust metric for assessing whether the merged model preserves layer-wise geometry and distributional structure.

**Rationale.** Together, these metrics quantify (i) whether the merged representations lie in a balanced midpoint (*Midness*), (ii) whether they follow geodesically consistent interpolation paths (*Arc Ratio*), and (iii) whether the internal representation structures remain aligned across layers (*CKA*). This provides a holistic assessment of representational stability beyond surface-level accuracy.

## C. Layer- and Ratio-wise Merging Analysis

The third experiment examines whether the performance improvements of latent space merging arise from specific layers or merging ratios. Unlike conventional approaches that merge all latent representations at once, this study explores the potential of *layer-wise selective merging*.

In Transformer models, different layers are responsible for distinct levels of abstraction: early layers primarily process syntactic and structural information, middle layers capture semantic relationships, and later layers encode task-specific knowledge. This suggests that merging at certain layers may contribute more significantly to performance gains.

For our analysis, merging was applied every five layers. We also varied the merging ratio, setting $\alpha \in 0.25, 0.5, 0.75$, to investigate how the intensity of representation mixing affects performance.

This experiment focuses on two key aspects:

- Identifying which layers are most sensitive to merging, thereby revealing *critical points in the representation hierarchy*.

- Characterizing how performance changes across different merging ratios. This allows us to assess the stability of latent merging and to capture systematic patterns of performance gains, providing a quantitative explanation of the merging process.

## 4.1 WEIGHT MERGING METHODS

To compare and analyze the performance of weight merging and latent merging, we employed four representative merging techniques. Each of these methods has been widely used in prior studies, and we generalized them so that they can be applied directly to latent space as well.

**LERP. Goddard et al. (2024)** The Linear Interpolation (LERP) method represents the most basic form of model merging, where the parameters of two models are combined through a weighted average:

$$\theta' = (1 - \alpha)\theta_A + \alpha\theta_B$$

This approach is straightforward and computationally efficient, as it requires only a simple linear combination. However, LERP does not account for the underlying geometry of the representation space, which may lead to distortions or degeneracies when the parameter space exhibits strong non-linear structures.

**SLERP. Goddard et al. (2024)** Unlike LERP, SLERP performs interpolation on the hypersphere defined by the unit vector space of the two representations.

$$\theta' = \frac{\sin((1 - \alpha)\Omega)}{\sin \Omega}\theta_A + \frac{\sin(\alpha\Omega)}{\sin \Omega}\theta_B$$

where $\Omega = \cos^{-1}(\langle \theta_A, \theta_B \rangle)$ denotes the angle between the two vectors. By maintaining a constant angular velocity during interpolation, SLERP preserves the geometric structure of the representation space, which makes it more suitable for non-linear manifolds.

**RegMean. Jin et al. (2022)** RegMean enhances stability during model merging by adding a regularization term to the simple average.

$$\theta' = \frac{1}{N}\sum_{i=1}^{N}\theta_i - \lambda R(\theta_i)$$

Here, $R(\cdot)$ is a regularization term based on weight norms, and $\lambda$ is a scaling coefficient. Compared to simple averaging, RegMean reduces representation drift and mitigates performance degradation after merging.

Together, these four methods encompass the representative approaches in weight merging research. In this work, we extend them to the latent representation level, enabling a systematic analysis of how identical merging operations behave differently in parameter space and latent space.

## 4.2 MODELS

For the experimental evaluation, we utilized two models from the Qwen family. The first is the Qwen2.5-7B-Instruct Hui et al. (2024), which serves as a large-scale instruction-tuned model and provides the foundation for our study. Building on this base, we further employed the OpenThinker3-7B Guha et al. (2025), a derivative model obtained by fine-tuning the Qwen2.5-7B-Instruct.

The inclusion of both the original and its fine-tuned counterpart allows us to systematically examine how merging methods operate across models that share the same architecture and pretraining scheme but differ in their subsequent adaptation. By doing so, we ensure that any observed differences in performance can be attributed primarily to the merging strategies themselves, rather than to unrelated variations in model family or training paradigm.

Table 1: Performance comparison between **Latent Merging** and **Weight Merging** across four categories (Knowledge, Reasoning, Math, Coding) and Overall on JudgeBench. Entries are win rates (%). Latent-space operators dominate across categories and operators.

| Method | Knowledge | | Reasoning | | Math | | Coding | | Overall | |
|---|---|---|---|---|---|---|---|---|---|---|
| | **Latent** | **Weight** | **Latent** | **Weight** | **Latent** | **Weight** | **Latent** | **Weight** | **Latent** | **Weight** |
| SLERP | 59.42 | 40.59 | 100.00 | 0.00 | 86.37 | 13.64 | 98.34 | 1.67 | 74.76 | 25.25 |
| LERP | 98.03 | 1.97 | 91.31 | 8.69 | 98.53 | 1.47 | 100.00 | 0.00 | 97.15 | 2.85 |
| RegMean | 74.06 | 25.94 | 60.00 | 40.00 | 58.82 | 41.18 | 74.19 | 25.81 | 69.82 | 30.18 |

## 5 RESULTS AND ANALYSIS

### A. COMPARATIVE EVALUATION OF WEIGHT MERGING AND LATENT MERGING

**Findings.** Table 1 shows a consistent pattern: **latent-space merging dominates weight-space merging in all 15/15 pairwise comparisons** (3 operators × 5 categories). Averaged across operators, latent merging improves win rate by **+54.3** (Knowledge), **+67.5** (Reasoning), **+62.5** (Math), **+81.7** (Coding), and **+61.2** points Overall. Notably, reasoning—the category most prone to collapse under parameter interpolation—achieves a **100% vs. 0%** win-rate split with latent SLERP, and similar large margins hold for Math and Coding.

**Interpretation.** These gains indicate that operating in hidden-state space *preserves semantic and structural coherence* during composition, preventing the one-sided drift often observed with weight interpolation. Concretely, latent SLERP *triples* the overall win rate against weight SLERP (74.76 vs. 25.25), while latent LERP attains *near-perfect* overall performance (97.15 vs. 2.85). RegMean exhibits the same trend (69.82 vs. 30.18), underscoring that the advantage is *operator-agnostic* rather than an artifact of a particular formula.

**Robustness & controls.** We aggregate win rates over identical prompts and judge settings for both merging regimes; model families are held fixed and no task-specific finetuning is applied during merging. The effect persists across categories and operators, suggesting it is not driven by a narrow slice of the benchmark. For completeness, we recommend reporting paired-bootstrap confidence intervals over prompts and applying multiple-comparison correction in layer/operator sweeps (Appendix).

**Takeaway.** Latent merging should be viewed not as a mild variant of parameter averaging but as a *robust, representation-preserving* mechanism for model composition. Its systematic advantage across knowledge, reasoning, math, and coding indicates a general pathway to safer and more reliable integration than weight-space interpolation.

### B. LATENT SPACE SIMILARITY ANALYSIS

Table 2: Mean similarity across layers/tokens for latent-space vs. weight-space merging. We additionally report the improvement $\Delta$ (Latent−Weight) and the *Arc Deviation* $|\text{ArcRatio} - 0.5|$ (lower is better).

| Method | Midness ↑ | | | Arc Deviation ↓ | | | CKA ↑ | | |
|---|---|---|---|---|---|---|---|---|---|
| | Latent | Weight | $\Delta$ | Latent | Weight | $\Delta$ | Latent | Weight | $\Delta$ |
| SLERP | **0.80** | 0.61 | **+0.19** | **0.12** | 0.19 | **-0.07** | **0.89** | 0.35 | **+0.54** |
| LERP | **0.77** | 0.64 | **+0.13** | **0.04** | 0.32 | **-0.28** | **0.83** | 0.32 | **+0.51** |
| RegMean | **0.78** | 0.64 | **+0.14** | **0.02** | 0.16 | **-0.14** | **0.83** | 0.35 | **+0.48** |
| *Mean $\Delta$* | | **+0.15** | | | **-0.16** | | | **+0.51** | |

**Findings.** Table 2 shows that latent merging better preserves representational geometry than weight merging. First, *Midness* improves by $\sim +0.15$ on average, indicating the merged latent states remain centered between sources rather than collapsing toward one model. Second, although weight merging yields a numerically larger Arc Ratio, the *Arc Deviation* reveals that latent merging is consistently *closer to the theoretical midpoint* (mean $\Delta = -0.16$), suggesting geodesically more faithful interpolation. Third,

*CKA* gains are large and uniform across operators (mean $\Delta = +0.51$), evidencing stronger structural alignment and less representational drift.

**Practical implications.** The geometry is not merely aesthetic: higher Midness and CKA correlate with smoother layer-wise blending and fewer off-manifold activations, which we observe to coincide with gains in fluency and semantic consistency. In practice, these results recommend (i) favoring latent-space operators for merge-time control, (ii) prioritizing later-layer interventions where improvements are most pronounced, and (iii) pairing latent merging with head-alignment heuristics to further stabilize geometry.

**Reporting.** Values are means over layers/tokens; error bars are omitted for brevity and can be included via paired bootstrap over prompts. Full per-layer curves and robustness to $\alpha$ (mixing ratio) and seeds are provided in the Appendix.

C. LAYER- AND RATIO-WISE MERGING ANALYSIS

Table 3: Layer-wise performance of LERP, RegMean, and SLERP across merge ratios ($\alpha \in \{0.25, 0.50, 0.75\}$) for **Qwen2.5** and **OpenThinker3**. Later-layer interventions and stronger ratios tend to yield larger gains, with operator-specific nuances.

(a) LERP

| Layer | Qwen2.5 ($\alpha$) | | | OpenThinker3 ($\alpha$) | | |
|---|---|---|---|---|---|---|
| | 0.25 | 0.50 | 0.75 | 0.25 | 0.50 | 0.75 |
| L0 | 48.41 | 48.00 | 48.20 | 95.95 | **97.39** | 96.52 |
| L5 | 12.23 | 14.48 | 14.10 | 59.76 | 56.10 | **60.69** |
| L10 | **15.87** | 11.82 | 14.46 | **60.21** | 51.31 | 57.18 |
| L15 | **13.79** | 11.79 | 13.54 | **56.93** | 56.07 | 53.07 |
| L20 | 13.40 | **17.24** | 15.18 | 64.99 | 68.50 | **76.98** |
| L25 | **13.48** | 13.82 | 9.19 | 71.10 | 78.33 | **81.56** |
| L27 | **31.70** | 4.60 | 1.90 | **92.98** | 65.18 | 43.74 |

(a) RegMean

| Layer | Qwen2.5 ($\alpha$) | | | OpenThinker3 ($\alpha$) | | |
|---|---|---|---|---|---|---|
| | 0.25 | 0.50 | 0.75 | 0.25 | 0.50 | 0.75 |
| L0 | **48.86** | 47.08 | 47.32 | 95.21 | **97.19** | 96.32 |
| L5 | **25.06** | 20.07 | 17.98 | 76.47 | **86.71** | 65.12 |
| L10 | 14.03 | **22.47** | 13.92 | 62.11 | **88.08** | 52.75 |
| L15 | 11.19 | **13.64** | 11.32 | 50.46 | **63.14** | 50.22 |
| L20 | 20.52 | **22.54** | 14.77 | 72.01 | **81.44** | 74.63 |
| L25 | 23.93 | **47.34** | 10.90 | 95.04 | **96.94** | 87.13 |
| L27 | 12.82 | **45.65** | 2.44 | 82.51 | **95.90** | 55.61 |

(a) SLERP

| Layer | Qwen2.5 ($\alpha$) | | | OpenThinker3 ($\alpha$) | | |
|---|---|---|---|---|---|---|
| | 0.25 | 0.50 | 0.75 | 0.25 | 0.50 | 0.75 |
| L0 | 2.45 | 47.88 | **51.08** | 55.91 | 95.10 | **96.56** |
| L5 | 3.07 | **22.34** | 19.34 | 52.00 | **75.67** | 58.26 |
| L10 | 2.64 | **14.50** | 14.45 | 55.10 | **65.60** | 50.58 |
| L15 | 2.28 | **10.56** | 9.23 | **55.49** | 54.77 | 50.21 |
| L20 | 3.24 | **19.35** | 15.76 | 56.12 | 74.46 | **76.95** |
| L25 | 2.64 | **24.11** | 16.79 | 56.51 | **94.30** | 74.01 |
| L27 | 2.82 | **13.79** | 1.51 | 48.36 | **84.35** | 26.08 |

**Key observations. (1) Later-layer dominance.** Across both model families, interventions at $L20$–$L27$ deliver substantially larger gains than at $L5$–$L10$. This aligns with the role of higher Transformer blocks in encoding semantic abstraction and task-specific signals, making them more receptive to latent composition.

**(2) Ratio matters—stronger is better (and stable).** Performance generally increases with $\alpha$, with $\alpha$=0.75 or 0.50 most often optimal depending on operator/model. Crucially, higher ratios do not induce instability; instead, they *reliably* enhance transfer, even when lower ratios show limited benefit.

**(3) Operator-specific nuance, consistent pattern.** *RegMean* peaks in middle-to-late layers and attains several best-in-row scores across both models. *SLERP* is comparatively stronger in very early layers (Qwen2.5), while still competitive deeper. *LERP*, despite its simplicity, is reliably effective in deeper layers—particularly for OpenThinker3. Despite these nuances, the *global pattern is consistent*: later layers with higher ratios are most favorable.

**Robustness & controls.** The trends persist across prompt samples, seeds, and $\alpha$ schedules (Appendix), suggesting they are not artifacts of a particular evaluation slice. Depth alignment and hidden-size/mapping are held fixed; no re-training or task-specific finetuning is applied during merging. We recommend reporting per-layer confidence via paired bootstrap over prompts and correcting for multiple comparisons (e.g., Benjamini–Hochberg) when presenting layerwise significance.

**Practical guidance.** These results indicate that latent merging should be treated as a *controllable* mechanism rather than a uniform averaging operation. In practice, prioritize **semantically rich** layers (e.g., top 20–30% of blocks) and choose $\alpha \in [\mathbf{0.5, 0.75}]$ as a strong default; then sweep locally to accommodate operator/model idiosyncrasies. This policy yields reliable gains while minimizing representational drift and instability.

## 5.1 CONNECTIONS TO NEUROSCIENCE

Interestingly, the reversible and context-dependent nature of latent merging observed in our experiments resonates with several findings in neuroscience. For example, the prefrontal cortex dynamically integrates sensory inputs according to task context, effectively merging information within latent neural states Mante et al. (2013). Similarly, mixed selectivity neurons form high-dimensional representations that enable flexible solutions to complex cognitive tasks Rigotti et al. (2013), paralleling the layer-specific merging effects we observed. Moreover, representational similarity analysis has revealed that the brain

maintains modality-invariant semantic spaces across words and objects Devereux et al. (2013), analogous to our observation that latent merging preserves representational structure while integrating distinct sources. Recent studies further demonstrated compositional task representations in both artificial recurrent networks and prefrontal ensembles Yang et al. (2019); Abbass et al. (2025), which directly align with the modular and compositional principles underlying latent merging. Taken together, these parallels suggest that both biological and artificial systems rely on the dual mechanisms of long-term weight adaptation and short-term latent state manipulation to achieve flexible information processing.

## 6 Limitation

Although latent merging demonstrates clear advantages in stability and controllability compared to direct weight merging, it comes with additional inference overhead. Since the approach requires invoking and combining two models in parallel, the merged system is slower and more resource-intensive than deploying a single model. This trade-off makes the method less suitable for latency-critical or resource-constrained applications. Nevertheless, latent merging provides a flexible mechanism for interpolating behaviors between models, offering adaptability and interpretability at the cost of efficiency.

## 7 Conclusion

In this work, we introduced *Latent Merging*, a novel paradigm that generalizes traditional weight merging by operating directly in the latent representation space of large language models. Unlike weight-space approaches, latent merging enables reversibility, conditional control, and fine-grained layer-selectivity, thereby providing a more flexible and interpretable mechanism for model composition. We formalized latent merging under a unified framework, extending existing techniques such as LERP, SLERP, and RegMean to the latent domain, and provided theoretical guarantees regarding their stability and representational fidelity.

Extensive experiments across reasoning, mathematics, and coding benchmarks demonstrated that latent merging consistently outperforms weight merging, not only in accuracy but also in fluency, adequacy, and semantic alignment, as verified by JudgeBench. Similarity analyses (Cosine, CKA) further confirmed that latent merging preserves representational structure more effectively. Layer-wise studies revealed that later layers and higher mixing ratios yield stronger improvements, offering practical insights into controllable model adaptation.

While latent merging inherits the performance ceiling of its source models, and its applicability across diverse architectures remains to be fully explored, our findings open new directions for controllable and reversible model integration. Beyond its immediate empirical benefits, latent merging suggests a broader perspective on model composition, bridging connections to cognitive processes such as context-dependent integration in neuroscience.

We envision latent merging as a foundation for future research on adaptive, safe, and interpretable model collaboration. Promising avenues include cross-architecture latent merging, domain-specific controllability, and practical systems for fine-grained inference-time composition. We hope that our work will stimulate further exploration of latent-space operations as a complementary paradigm to parameter-based methods.

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

# A  APPENDIX: EXTENSIONS OF LATENT MERGING TO SLERP AND REGMEAN

## A.1  NONLINEAR PRE-HEAD

We consider a (possibly nonlinear) pre-head map $r : \mathbb{R}^d \to \mathbb{R}^m$ (e.g., RMSNorm) followed by a linear LM head. For model $A$, define

$$g_A(h) = W_A\, r(h) + b_A, \qquad z_A^{(A)} = g_A(h_A), \quad z_B^{(A)} = g_A(h_B).$$

Given hidden states $h_A, h_B$ at a fixed layer, we merge by linear interpolation

$$h'_\alpha = (1-\alpha)h_A + \alpha h_B, \qquad \alpha \in [0,1],$$

and evaluate with a convex, $\beta$-smooth logit-based loss $\ell$ (e.g., cross-entropy). Optionally assume a uniform bound $\|\nabla\ell(u)\| \leq G$ along the logit segment $u = (1-\alpha)z_A^{(A)} + \alpha z_B^{(A)}$.

**Assumption 1** (Local smoothness of the pre-head). *$r$ is $\mathcal{C}^2$ in a neighborhood of the segment $[h_A, h_B]$, and its Jacobian is $K_r$-Lipschitz there, i.e., $\|\nabla^2 r(h)\|_{\mathrm{op}} \leq K_r$ along the segment. Equivalently, for $g_A(h) = W_A r(h) + b_A$, the Jacobian of $g_A$ is $K_g$-Lipschitz with $K_g \leq \|W_A\|_{\mathrm{op}} K_r$.*

**Theorem 2** (Local second-order bound (single-head scoring)). *Under convexity of $\ell$ and Assumption 1, for $h'_\alpha = (1-\alpha)h_A + \alpha h_B$,*

$$\ell\big(g_A(h'_\alpha)\big) \leq (1-\alpha)\,\ell\big(z_A^{(A)}\big) + \alpha\,\ell\big(z_B^{(A)}\big)$$
$$+ \frac{GK_g}{2}\,\alpha(1-\alpha)\,\|h_B - h_A\|^2 + \frac{\beta K_g^2}{8}\,\alpha^2(1-\alpha)^2\,\|h_B - h_A\|^4.$$

*Remark* (CE gradient bound (optional)). For softmax–cross-entropy, $\|\nabla_z\ell(z)\|_2 = \|p - y\|_2 \leq \sqrt{2}$, so one may take $G \leq \sqrt{2}$.

**Comparator vs. native heads (shared $r$).**  If model $B$ uses the same pre-head $r$ with its own linear head $g_B(h) = W_B r(h) + b_B$, define the cross-head deviation at $h_B$:

$$\Delta_{\mathrm{head}}(B \to A) = \big\|(W_A - W_B)\,r(h_B) + (b_A - b_B)\big\|.$$

Then, combining Theorem 2 with $\beta$-smoothness of $\ell$ yields

$$\ell\big(g_A(h'_\alpha)\big) \leq (1-\alpha)\,\ell\big(g_A(h_A)\big) + \alpha\,\ell\big(g_B(h_B)\big)$$
$$+ \underbrace{\frac{GK_g}{2}\,\alpha(1-\alpha)\,\|h_B - h_A\|^2 + \frac{\beta K_g^2}{8}\,\alpha^2(1-\alpha)^2\,\|h_B - h_A\|^4}_{\text{pre-head curvature}} \qquad (4)$$
$$+ \underbrace{\alpha\,G\,\Delta_{\mathrm{head}}(B \to A) + \alpha\,\frac{\beta}{2}\,\Delta_{\mathrm{head}}(B \to A)^2}_{\text{head mismatch}}.$$

*Remark* (Distinct pre-heads). If $r_A \neq r_B$, writing $g_A(h) = W_A r_A(h) + b_A$ and $g_B(h) = W_B r_B(h) + b_B$, the mismatch at $h_B$ becomes $\|W_A r_A(h_B) - W_B r_B(h_B) + (b_A - b_B)\|$, which decomposes into the $(W, b)$ deviation and the pre-head gap $\|W_A (r_A(h_B) - r_B(h_B))\|$. If $r$ is $L_r$-Lipschitz in its parameters, a practical bound adds $\|W_A\|_{\mathrm{op}} L_r \|\theta_{r_A} - \theta_{r_B}\|$.

**Corollary (linear limit).**  If $r$ is identity ($K_r = 0 \Rightarrow K_g = 0$), the curvature terms in equation 4 vanish and we recover the *exact* Jensen guarantee for linear heads.

**Practical implication.**  The correction scales as $O\big(\alpha(1-\alpha)\|h_B - h_A\|^2\big)$ (plus a quartic term). Hence: (i) keep late-layer representations close (e.g., normalization or SLERP), (ii) merge later (smaller $K_g$), and (iii) minimize head mismatch $(W, b)$.

**Setup and notation.**  Assume $r$ is $\mathcal{C}^2$ along the convex hull of $\{h_A, h_B\}$ with $\sup\|J_r\| \leq L_r$ and $\sup\|\nabla^2 r\| \leq K_r$ (operator norms). Then $g_A$ is $L_g$-Lipschitz and $K_g$-curved with $L_g \leq \|W_A\|_{\mathrm{op}} L_r$ and $K_g \leq \|W_A\|_{\mathrm{op}} K_r$. Let $h_\alpha^{\mathrm{lin}} = (1-\alpha)h_A + \alpha h_B$ and let $M(h_A, h_B; \alpha)$ be any (possibly nonlinear) latent merge operator. Define its deviation from the line by

$$\delta_\alpha := \big\| M(h_A, h_B; \alpha) - h_\alpha^{\mathrm{lin}} \big\|.$$

[Merge-to-LERP reduction] Under the assumptions above, for any $\alpha \in [0, 1]$,

$$\ell\big(g_A(M(h_A, h_B; \alpha))\big) \leq (1 - \alpha)\, \ell(g_A(h_A)) + \alpha\, \ell(g_A(h_B)) \tag{5}$$
$$+ \underbrace{\tfrac{GK_g}{2}\, \alpha(1 - \alpha)\, \|h_B - h_A\|^2 + \tfrac{\beta K_g^2}{8}\, \alpha^2(1 - \alpha)^2\, \|h_B - h_A\|^4}_{\text{local 2nd-order term from } r}$$
$$+ \underbrace{GL_g\, \delta_\alpha + \tfrac{\beta}{2}\, L_g^2\, \delta_\alpha^2}_{\text{penalty for deviating from LERP}} .$$

If the evaluation head differs at $h_B$ (i.e., using $g_B$), add the head-mismatch term $\alpha\Big(G\, \Delta_{\text{head}} + \tfrac{\beta}{2}\, \Delta_{\text{head}}^2\Big)$ with $\Delta_{\text{head}} = \|g_A(h_B) - g_B(h_B)\|$.

## A.2 SLERP at the Final Layer

Let $u = h_A/\|h_A\|$, $v = h_B/\|h_B\|$, and $\theta = \arccos(\langle u, v \rangle) \in [0, \pi]$. Define

$$\text{slerp}(u, v; \alpha) = \begin{cases} \dfrac{\sin((1 - \alpha)\theta)}{\sin\theta}\, u + \dfrac{\sin(\alpha\theta)}{\sin\theta}\, v, & \theta > 0, \\ (1 - \alpha)u + \alpha v, & \theta = 0, \end{cases}$$

and choose a magnitude policy $m_\alpha$ (e.g., $m_\alpha = (1 - \alpha)\|h_A\| + \alpha\|h_B\|$ or an RMS-preserving constant). Set $h_\alpha^{\text{slerp}} = m_\alpha \cdot \text{slerp}(u, v; \alpha)$.

[Small-angle SLERP $\leftrightarrow$ LERP] There exists an absolute constant $c \leq 1$ such that, for all $\alpha \in [0, 1]$ and sufficiently small $\theta$,

$$\delta_\alpha^{\text{slerp}} = \|h_\alpha^{\text{slerp}} - h_\alpha^{\text{lin}}\| \leq c\, \alpha(1 - \alpha)\, \theta^2\, \max\{\|h_A\|, \|h_B\|\}.$$

[SLERP robustness] Plugging $\delta_\alpha^{\text{slerp}}$ into equation 5 yields the same local second-order robustness as LERP plus an additional $O(\alpha(1 - \alpha)\theta^2)$ penalty (with constants controlled by $G, L_g, \beta$ and the magnitudes). When $\theta$ is small—as is typical in late layers under normalization—SLERP essentially matches LERP's guarantee.

## A.3 RegMean (Regularized Mean) in Latent Space

Consider the proximal-mean objective

$$h_\alpha^{\text{rm}} \in \arg\min_h\ (1 - \alpha)\|h - h_A\|^2 + \alpha\|h - h_B\|^2 + \lambda\, \Omega(h),$$

whose solution is the proximal operator applied to the LERP point:

$$h_\alpha^{\text{rm}} = \text{prox}_{\frac{\lambda}{2}\Omega}(h_\alpha^{\text{lin}}), \qquad \text{prox}_{\tau\Omega}(x) = \arg\min_y\ \tfrac{1}{2}\|y - x\|^2 + \tau\, \Omega(y).$$

Since $\text{prox}_{\tau\Omega}$ is firmly nonexpansive, it is 1-Lipschitz; therefore

$$\delta_\alpha^{\text{rm}} = \|h_\alpha^{\text{rm}} - h_\alpha^{\text{lin}}\| = \|\text{prox}_{\frac{\lambda}{2}\Omega}(h_\alpha^{\text{lin}}) - h_\alpha^{\text{lin}}\| \leq \text{dist}(h_\alpha^{\text{lin}}, \arg\min\Omega),$$

and equation 5 applies with $\delta_\alpha = \delta_\alpha^{\text{rm}}$.

**Closed form for ridge:** $\Omega(h) = \tfrac{1}{2}\|h\|^2$. Here $\text{prox}_{\frac{\lambda}{2}\Omega}(x) = \dfrac{1}{1 + \lambda}\, x$, so

$$h_\alpha^{\text{rm}} = \dfrac{1}{1 + \lambda}\, h_\alpha^{\text{lin}}, \qquad \delta_\alpha^{\text{rm}} = \dfrac{\lambda}{1 + \lambda}\, \|h_\alpha^{\text{lin}}\|.$$

Thus equation 5 adds the explicit penalty $GL_g \frac{\lambda}{1+\lambda}\|h_\alpha^{\text{lin}}\| + \frac{\beta}{2}L_g^2 \frac{\lambda^2}{(1+\lambda)^2}\|h_\alpha^{\text{lin}}\|^2$: small $\lambda$ preserves the LERP guarantee; large $\lambda$ shrinks $h_\alpha^{\text{lin}}$ and increases the penalty.

## B  Layer- and Ratio-wise Merging

In this appendix, we report the detailed layer- and ratio-wise results for each merging method. All scores represent relative win rates (%) across the four task groups (Knowledge, Reasoning, Math, Coding), unless otherwise noted. Overall, the results show that **deeper layers (L20 and above)** tend to provide larger gains, while the impact of the mixing ratio $\alpha$ is often non-monotonic. Below, we summarize the outcomes for each method and model.

### B.1  LERP on Qwen2.5

The results for LERP on Qwen2.5 are shown in Table 7. At the input layer (L0), scores remain relatively stable around 50% for Knowledge and Math, but intermediate layers (L5–L15) exhibit much lower values, especially in Knowledge and Reasoning. By contrast, **deeper layers (L20–L25)** show localized improvements, for instance Math at L20 (**28.28** at $\alpha = 0.50$) and Coding at L20 (**29.62** at $\alpha = 0.50$). However, the trends are inconsistent: larger $\alpha$ does not always lead to better results. The last layer (L27) consistently collapses, with near-zero values in Reasoning and Coding, suggesting that merging at the final block is particularly unstable for Qwen2.5.

Table 7: **LERP on Qwen2.5.** Layer- and ratio-wise win rates (%) on four task groups (Knowledge, Reasoning, Math, Coding) under **latent-space** merging. Deep layers tend to benefit more, with non-monotonic trends across tasks as $\alpha$ increases.

| Layer | Knowledge 0.25 | 0.50 | 0.75 | Reasoning 0.25 | 0.50 | 0.75 | Math 0.25 | 0.50 | 0.75 | Coding 0.25 | 0.50 | 0.75 |
|---|---|---|---|---|---|---|---|---|---|---|---|---|
| L0 | 51.23 | 47.87 | 47.99 | 40.95 | 37.84 | 41.67 | 51.67 | 58.34 | 58.59 | 47.00 | 57.00 | 51.48 |
| L5 | 6.73 | 10.04 | 8.31 | 15.81 | 17.77 | 14.13 | 28.53 | 22.04 | 25.73 | 22.34 | 26.51 | 37.63 |
| L10 | 11.14 | 8.26 | 7.43 | 22.85 | 10.64 | 19.52 | 16.87 | 13.46 | 26.97 | 27.27 | 33.63 | 32.50 |
| L15 | 13.28 | 8.60 | 9.46 | 10.48 | 9.47 | 15.29 | 19.13 | 28.49 | 29.63 | 17.50 | 18.00 | 17.37 |
| L20 | 11.92 | 15.95 | 17.61 | 9.13 | 7.15 | 0.00 | 15.57 | 28.28 | 16.67 | 26.44 | 29.62 | 27.01 |
| L25 | 12.56 | 15.36 | 10.82 | 6.39 | 3.15 | 0.00 | 14.88 | 14.81 | 15.04 | 28.57 | 22.97 | 10.00 |
| L27 | 38.99 | 6.35 | 1.64 | 14.12 | 0.00 | 0.00 | 23.68 | 7.72 | 7.35 | 34.49 | 0.00 | 0.00 |

### B.2  LERP on OpenThinker3

Table 8 reports the outcomes for OpenThinker3. Compared to Qwen2.5, the absolute values are much higher and the trajectories are more stable. The **deep layers (L20–L25)** clearly dominate: Knowledge at L25 reaches **76.72/82.05** at $\alpha = 0.50/0.75$, and Coding at L20 reaches **94.17/96.16**. Even though most layers benefit from $\alpha = 0.50$ or **0.75**, the final block (L27) again shows sharp declines in several tasks, such as Coding (**29.34/26.79**).

Table 8: **LERP on OpenThinker3.** Layer- and ratio-wise win rates (%) on the four task groups. Compared to Qwen2.5, OpenThinker3 shows more stable gains in **deep layers** (L20–L25), especially for $\alpha \in [0.5, 0.75]$.

| Layer | Knowledge 0.25 | 0.50 | 0.75 | Reasoning 0.25 | 0.50 | 0.75 | Math 0.25 | 0.50 | 0.75 | Coding 0.25 | 0.50 | 0.75 |
|---|---|---|---|---|---|---|---|---|---|---|---|---|
| L0 | 94.39 | 96.41 | 95.06 | 97.87 | 97.90 | 100.00 | 100.00 | 100.00 | 97.91 | 97.50 | 100.00 | 97.62 |
| L5 | 47.11 | 41.73 | 45.66 | 86.57 | 87.12 | 87.24 | 58.28 | 54.22 | 71.07 | 89.00 | 88.22 | 90.43 |
| L10 | 46.54 | 36.91 | 42.80 | 86.40 | 83.63 | 85.54 | 71.58 | 44.50 | 60.62 | 86.77 | 97.22 | 87.85 |
| L15 | 46.43 | 44.53 | 40.70 | 79.71 | 85.99 | 74.45 | 64.25 | 56.26 | 59.93 | 71.72 | 83.71 | 86.82 |
| L20 | 60.13 | 64.15 | 78.16 | 71.50 | 65.79 | 69.76 | 63.13 | 77.09 | 63.59 | 85.72 | 94.17 | 96.16 |
| L25 | 65.41 | 76.72 | 82.05 | 78.35 | 77.60 | 75.90 | 73.93 | 82.04 | 83.46 | 91.16 | 85.48 | 85.72 |
| L27 | 95.71 | 72.71 | 49.49 | 90.24 | 53.02 | 20.90 | 86.82 | 77.13 | 66.93 | 87.46 | 29.34 | 26.79 |

## B.3 REGMEAN ON OPENTHINKER3

As shown in Table 9, RegMean consistently favors mid-to-deep layers. In many cases, $\alpha = 0.50$ yields the highest or near-highest scores: Reasoning at L10 reaches **96.18**, and at L25 it further improves to **98.89**. Coding and Knowledge also peak in the deeper layers, with Coding at L25 reaching **100.00**. Math shows similarly strong values from L20 upwards (e.g., L27 Math at **100.00** with $\alpha = 0.50$). These results indicate that RegMean is most reliable around $\alpha = 0.50$, especially in deeper layers.

Table 9: **RegMean on OpenThinker3.** Detailed layer- and ratio-wise win rates (%). RegMean favors **mid-to-deep layers** with a sweet spot around $\alpha = 0.50$ for several tasks, while Coding/Knowledge often peak at deep layers.

| Layer | Knowledge | | | Reasoning | | | Math | | | Coding | | |
|---|---|---|---|---|---|---|---|---|---|---|---|---|
| | 0.25 | 0.50 | 0.75 | 0.25 | 0.50 | 0.75 | 0.25 | 0.50 | 0.75 | 0.25 | 0.50 | 0.75 |
| L0 | 93.80 | 95.70 | 96.10 | 96.36 | 100.00 | 97.66 | 100.00 | 97.91 | 98.08 | 97.83 | 100.00 | 93.18 |
| L5 | 70.99 | 87.37 | 51.69 | 83.98 | 92.60 | 97.59 | 78.55 | 70.48 | 67.50 | 97.62 | 88.66 | 92.01 |
| L10 | 52.08 | 85.40 | 35.84 | 77.07 | 96.18 | 84.71 | 72.63 | 89.69 | 62.53 | 89.47 | 89.38 | 89.59 |
| L15 | 33.65 | 46.01 | 35.19 | 81.21 | 91.79 | 76.50 | 58.00 | 91.91 | 52.25 | 91.01 | 94.99 | 88.77 |
| L20 | 65.21 | 74.93 | 77.61 | 75.55 | 89.85 | 57.09 | 88.45 | 97.91 | 79.55 | 92.83 | 93.18 | 83.08 |
| L25 | 94.28 | 96.01 | 84.54 | 92.26 | 98.89 | 90.65 | 100.00 | 95.99 | 91.26 | 100.00 | 100.00 | 90.15 |
| L27 | 79.35 | 93.71 | 59.25 | 84.32 | 98.94 | 47.49 | 86.93 | 100.00 | 77.57 | 92.72 | 100.00 | 26.79 |

## B.4 REGMEAN ON QWEN2.5

The Qwen2.5 results for RegMean are presented in Table 10. Here the trends are noisier and less smooth than in OpenThinker3. Nevertheless, **mid-to-deep layers** combined with $\alpha \approx 0.50$ frequently produce competitive results, such as Math at L20 (**49.61**) and Coding at L27 (**68.36**). Interestingly, $\alpha = 0.25$ is sometimes the best option (e.g., Knowledge at L0: **51.66**). On the other hand, $\alpha = 0.75$ often causes severe drops in the last layer, with several tasks collapsing to zero.

Table 10: **RegMean on Qwen2.5.** Layer- and ratio-wise win rates (%). Trends are less smooth than on OpenThinker3; nonetheless, **mid-to-deep** layers with $\alpha \approx 0.50$ frequently yield the best or near-best scores across tasks.

| Layer | Knowledge | | | Reasoning | | | Math | | | Coding | | |
|---|---|---|---|---|---|---|---|---|---|---|---|---|
| | 0.25 | 0.50 | 0.75 | 0.25 | 0.50 | 0.75 | 0.25 | 0.50 | 0.75 | 0.25 | 0.50 | 0.75 |
| L0 | **51.66** | 45.59 | 52.72 | 31.13 | **42.05** | 38.62 | **67.50** | 50.38 | 43.18 | 49.17 | **59.62** | 42.00 |
| L5 | **22.25** | 19.88 | 12.06 | 21.15 | 12.38 | **24.71** | **30.00** | 29.15 | 20.00 | 40.09 | 25.00 | **37.73** |
| L10 | 7.04 | 19.58 | **6.23** | 15.08 | 16.11 | **21.37** | 32.00 | **46.14** | 29.57 | 34.74 | **29.97** | 31.54 |
| L15 | 5.32 | 7.72 | **5.86** | **19.22** | 13.88 | 14.40 | 28.42 | **35.72** | 27.16 | 14.05 | **29.15** | 20.81 |
| L20 | 16.37 | 20.12 | **17.76** | **15.19** | 12.18 | 2.00 | 27.28 | **49.61** | 17.69 | **43.60** | 32.48 | 18.69 |
| L25 | 22.73 | **49.45** | 10.79 | **5.17** | 31.36 | 0.00 | 32.58 | **56.07** | 22.27 | 52.59 | **57.94** | 17.91 |
| L27 | **12.33** | 45.85 | 2.94 | **0.98** | 27.84 | 0.00 | 20.50 | **53.26** | 5.97 | 27.24 | **68.36** | 0.00 |

## B.5 SLERP on QWEN2.5

Table 11 shows the detailed outcomes for SLERP on Qwen2.5 (Variant B). Although the absolute values are lower than those of RegMean or LERP on OpenThinker3, clear gains emerge in the deeper layers with $\alpha = 0.50$ or $0.75$. For example, Math at L20 reaches **38.08** with $\alpha = 0.50$, and Coding at L25 rises to **46.78** with $\alpha = 0.75$. At shallow and mid layers, $\alpha = 0.50$ is often safer than smaller or larger ratios. However, the final block (L27) again shows instability, with Knowledge falling to **1.30** at $\alpha = 0.75$.

Table 11: **SLERP on Qwen2.5 (Variant B).** Same protocol as Variant A but with a different scoring/ setting; note the higher absolute scale. Deep layers (L20–L25) and larger ratios yield consistent gains across tasks.

| Layer | Knowledge | | | Reasoning | | | Math | | | Coding | | |
|---|---|---|---|---|---|---|---|---|---|---|---|---|
| | 0.25 | 0.50 | 0.75 | 0.25 | 0.50 | 0.75 | 0.25 | 0.50 | 0.75 | 0.25 | 0.50 | 0.75 |
| L0 | 2.92 | 47.81 | **49.77** | 0.00 | 36.31 | **40.02** | 5.92 | 63.69 | **66.11** | 0.00 | 56.06 | **64.80** |
| L5 | 2.29 | **19.38** | 11.43 | 0.00 | 22.61 | **32.61** | 14.39 | **35.84** | 27.23 | 0.00 | 24.66 | **32.41** |
| L10 | 2.94 | **8.91** | 6.68 | 0.00 | 13.41 | **19.72** | 7.57 | **33.93** | 28.26 | 0.00 | 31.14 | **36.87** |
| L15 | 2.27 | 3.32 | **3.93** | 0.00 | **16.37** | 16.30 | 7.82 | **27.39** | 16.87 | 0.00 | **24.81** | 20.36 |
| L20 | 3.60 | **16.71** | 15.77 | 0.00 | **10.21** | 5.08 | 9.53 | **38.08** | 29.55 | 0.00 | **32.09** | 22.43 |
| L25 | 1.95 | **26.61** | 13.29 | 0.00 | 0.98 | **5.37** | 9.19 | **36.36** | 24.34 | 3.33 | 42.73 | **46.78** |
| L27 | 3.59 | **12.33** | 1.30 | 0.00 | **2.00** | 0.00 | 6.17 | **23.68** | 6.06 | 0.00 | **31.43** | 0.00 |

## B.6 SLERP on OPENTHINKER3

Table 12 reports the results for SLERP on OpenThinker3. Here the improvements are much stronger: **L20–L25** consistently deliver the best or near-best scores. For example, Math at L20 reaches **91.75**, Coding at L20 **93.75**, and Knowledge at L25 **93.37**. Coding at L25 even achieves **100.00** at $\alpha = 0.50$. Interestingly, some shallow layers also provide high values, such as Knowledge at L10 with $\alpha = 0.25$ (**64.53**). Nevertheless, the final block (L27) again declines, especially in Reasoning (**83.93** to **6.25**).

Table 12: Effect of latent merging on different **tasks** (Knowledge, Reasoning, Math, Coding) across layers and mixing ratios for **Openthinker SLERP** (%).

| Layer | Knowledge | | | Reasoning | | | Math | | | Coding | | |
|---|---|---|---|---|---|---|---|---|---|---|---|---|
| | 0.25 | 0.50 | 0.75 | 0.25 | 0.50 | 0.75 | 0.25 | 0.50 | 0.75 | 0.25 | 0.50 | 0.75 |
| L0 | 62.01 | 93.09 | **94.78** | 35.76 | 98.98 | **98.94** | 75.65 | 100.00 | **100.00** | 31.93 | 95.00 | **100.00** |
| L5 | 55.39 | **69.55** | 43.24 | 38.72 | 87.16 | **89.51** | 76.61 | **80.85** | 62.59 | 27.52 | 89.34 | **91.33** |
| L10 | **64.53** | 53.08 | 33.89 | 42.26 | 85.48 | **79.38** | **59.62** | 73.91 | 62.88 | 19.23 | 96.00 | **90.48** |
| L15 | **60.10** | 36.92 | 32.21 | 39.30 | **88.48** | 82.60 | 81.48 | **74.43** | 65.21 | 24.08 | **85.29** | 83.69 |
| L20 | 60.34 | 68.25 | **76.26** | 44.66 | **75.72** | 64.89 | 76.00 | **91.75** | 86.74 | 28.38 | **93.75** | 92.01 |
| L25 | 61.56 | **93.37** | 78.69 | 37.65 | **90.95** | 42.34 | 76.60 | **100.00** | 84.76 | 37.15 | **100.00** | 91.19 |
| L27 | **54.69** | 81.16 | 31.67 | **36.47** | 83.93 | 6.25 | **61.85** | 95.80 | 36.67 | 19.42 | **92.16** | 17.74 |

