# OpenReview forum: "Latent Merging: Dynamic and Reversible Composition of Large Language Models"
_ICLR.cc/2026/Conference — ICLR 2026 Conference Withdrawn Submission_

### Official Review · Reviewer_48nT · 2025-10-22

**Soundness:** 2
**Presentation:** 2
**Contribution:** 2
**Rating:** 2
**Confidence:** 3

**Summary:**

The paper proposes Latent Merging, which merges models in representation space rather than parameter space, enabling dynamic and reversible control, and demonstrates  improvements over weight merging on JudgeBench

**Strengths:**

The paper proposes merging in representation space rather than parameter space, enabling dynamic and reversible model composition at inference time. It provides theoretical analysis that formalizes the representation alignment problem under RMSNorm nonlinearity and head mismatch.

**Weaknesses:**

1. Only pairwise comparisons available from JudgeBench LLM-as-Judge. Reliability Not Validated： JudgeBench provides only pairwise win rates, which cannot verify absolute performance improvement. For math and coding tasks with verifiable answers, the paper should report exact match accuracy. The 100% win rate for SLERP on reasoning is uninterpretable.
2.  While Section 4 and the results claim that later-layer interventions deliver substantially larger gains, Table 3 reveals catastrophic failure at the final layer L27 across all operators and both models,
3. Absence of baseline comparisons: Simple ensembling, where logits from both models are combined at the output layer, incurs the same 2× inference cost as latent merging but requires no hidden-state manipulation. Knowledge distillation could train a single student model to mimic both teachers, achieving a deployment cost of 1× rather than 2×, making it far more practical for production
4. The computational cost analysis is inadequate: The actual cost is substantial: double the memory footprint, doubled inference latency. The paper didn't justify when this 2× cost is worthwhile, what performance gain threshold makes it acceptable, or how it compares to alternatives at matched budgets, such as training a larger single model or using MoE Models.

**Questions:**

Raised in the Weaknesses section above

---

### Official Review · Reviewer_2Poq · 2025-10-25

**Soundness:** 2
**Presentation:** 2
**Contribution:** 2
**Rating:** 2
**Confidence:** 4

**Summary:**

The author proposes an idea of latent merging, which, by leveraging traditional merging methods, achieves better performance than weight merging. The author tests the advantages of latent merging on Qwen2.5-7B and its fine-tuned model, OpenThinker3, compared to direct weight merging. Additionally, the author provides a brief theoretical analysis to demonstrate the error control of the merge operation (under a simplified RMSnorm setting).

**Strengths:**

1.  Latent merging appears to be a more competitive method than weight merging in terms of performance.
2.  The author has provided a relatively complete analysis and background introduction.

**Weaknesses:**

I believe this paper has several issues that need to be addressed, which are of great concern to me.

1.  A significant advantage of weight merging is its ability to combine the strengths of two models while avoiding the need to compute the activations for both networks separately. However, latent merging does not seem to share this benefit. As I understand it, all experiments in this paper require loading both the Qwen2.5 and OpenThinker models simultaneously to obtain their latent activations for the merge. I am concerned that this approach will be difficult to scale up.
2.  The paper only compares the performance of weight merging and latent merging. I believe it is essential to include a baseline comparison with a control group (i.e., without any merging) to demonstrate that these merging methods are indeed meaningful.
3.  The writing in this paper is disastrous. My understanding is that all experiments are based on merging Qwen2.5 and OpenThinker3, which are two large models from the same family. However, the author fails to state this clearly. Consequently, I cannot understand why Table 3 contains two separate columns (Qwen and OpenThinker) within each sub-table. From my perspective, an $\alpha = 0.25$ for Qwen2.5 should imply a proportion of $1 - \alpha = 0.75$ for OpenThinker, yet the values in these two columns are completely different. Furthermore, I do not understand what "performance" the author is referring to in the caption, as the benchmark is not specified.

    The organization of the paper is also highly problematic. I believe that parts of Section 4 do not belong in the "Experimental Setup," such as sections 4.1 B, C, and the remaining content. This material should be merged directly with the results section. (The numbering scheme in the paper is also strange. For instance, in Section 4, the first three subsections are labeled A, B, and C, while the subsequent ones are numbered 4.1, 4.2, etc.)
4.  I believe the paper needs to include evaluations on common, fundamental benchmarks that do not rely on large model-based judgments, in addition to JudgeBench. I also think the paper should provide some examples or clarify the evaluation content and the results given by the judge model.
5.  Theorem 1 does not seem to provide any valuable information. The paper's control over the loss depends on the magnitude of $\Vert h_B - h_A \Vert$. However, in a weight merging scenario, this distance is unlikely to be small; otherwise, the merging would be meaningless.

**Questions:**

In Table 1, why does the simplest LERP method appear to perform so much better than more sophisticated methods like SLERP?

---

### Official Review · Reviewer_F72G · 2025-10-25

**Soundness:** 4
**Presentation:** 3
**Contribution:** 3
**Rating:** 4
**Confidence:** 4

**Summary:**

The paper introduces Latent Merging, a novel approach that combines large language models by merging their hidden representations rather than their weights. This allows dynamic, reversible, and layer-selective composition without altering model parameters. The authors provide theoretical analysis explaining when such merging is stable and demonstrate that it consistently outperforms traditional weight-merging across reasoning, math, and coding tasks. They also conduct several ablation studies that provide insights into the capabilities and behavior of latent merging.

**Strengths:**

The paper:
- Proposes a novel and well-motivated approach to combine LLMs by merging their latent representations instead of weights.

- Demonstrates clear empirical superiority of latent merging over traditional weight-merging methods across diverse reasoning and coding tasks.

- Includes comprehensive ablation studies that uncover valuable insights into when and how latent merging is most effective.

- Presents both theoretical analysis and practical guidelines, making the work impactful and reproducible.

**Weaknesses:**

The study:
- Lacks cross-family and cross-size model merging experiments, which would better demonstrate the generality and advantage of latent merging.

- Relies mainly on pairwise LLM-judge evaluations, which can introduce known biases and potentially inflate perceived gains; using universal metrics (e.g., accuracy) would strengthen the evaluation and better show the advantages of latent merging.

- Exhibits a narrow model scope and limited task diversity, focusing only on a single model family and a few benchmark categories.

- Missing a quantitative cost or latency analysis, which would be valuable for assessing the practical feasibility of deploying latent merging at scale.

**Questions:**

- What would be the impact of latent merging on other LLM behaviors such as hallucination or factual consistency?

- Since order bias is a known issue in pairwise LLM judging, did you randomize the order of latent vs weight merging outputs during evaluation to mitigate this effect?

- How might latent merging interact with domain adaptation or multi-task learning, especially when the merged models come from different fine-tuning objectives?

- Could the authors elaborate on whether layer-wise adaptive mixing ratios (instead of fixed α values) might further enhance performance or stability?

---

### Official Review · Reviewer_DVnc · 2025-10-29

**Soundness:** 3
**Presentation:** 2
**Contribution:** 2
**Rating:** 2
**Confidence:** 3

**Summary:**

This paper presents Latent Merging, a new paradigm for combining LLMs by dynamically interpolating their hidden states during inference, rather than statically merging their weights. This approach enables reversible, layer-specific control and is shown to dramatically outperform traditional weight merging across diverse benchmarks. The authors support their method with a theoretical framework and experiments, demonstrating preservation of the models' internal representational structure.

**Strengths:**

- The paper tackles the well-known instability and irreversibility of weight merging, not with a heuristic patch, but with a paradigm shift. By moving composition to the latent space, it addresses the source of the problem, global parameter modification, offering a more robust and theoretically sound solution.
- The work reframes model composition as a controllable, dynamic process. The ability to perform reversible, layer-specific interventions at inference time is a step forward, treating LLMs less as static artifacts and more as programmable computational graphs.
- The paper is generally easy to follow.

**Weaknesses:**

- The core limitation of inference overhead (doubling latency and memory) is noted but perhaps understated. This presents a substantial barrier to real-world adoption in resource-constrained or latency-critical applications, positioning the method more as a powerful analytical tool than a production-ready technique.
- The empirical validation is conducted on two models from the same family (Qwen2.5 and its finetune). This controlled setting leaves the method's effectiveness on architecturally diverse models (e.g., Llama + Mistral) or models with different tokenizers entirely unproven.
- Experiments are performed on 7B models. It is unclear how the memory and computational costs will scale to state-of-the-art large models (100B+ parameters), where duplicating the model graph for inference is often infeasible.
- The analysis reveals that performance is highly dependent on the choice of layers and mixing ratios (α), which vary by model pair and operator. This suggests that a potentially expensive search is required to find the optimal configuration for any new merge, reducing its "plug-and-play" simplicity.

**Questions:**

1. Regarding the significant inference overhead, have the authors explored sparser merging strategies (e.g., intervening only at a few critical layers identified via a heuristic) to find a better trade-off between performance and latency? Furthermore, could the desirable merged behavior be distilled back into a single, efficient model for deployment?
2. The experiments convincingly demonstrate merging within the Qwen family. What are the anticipated challenges of applying Latent Merging to models with different architectures, hidden state dimensions, or tokenizers? Would this necessitate trained projection layers, and how would that affect the "data-free" claim of the method?
3. The paper focuses on synergistic composition. How does the framework handle cases where models possess conflicting knowledge (e.g., different answers to a factual question) or opposing fine-tuning (e.g., one model is heavily safety-aligned, the other is not)? Does interpolation lead to a coherent compromise, dominance by one model, or unpredictable behavior?
4. The analysis reveals that performance is highly dependent on the choice of layer and mixing ratio. Beyond an empirical sweep, is there a more principled or efficient method to identify the optimal layers for merging? For instance, could a preliminary CKA analysis across layers predict which are most amenable to composition, thus reducing the search cost?

---

### Note · Authors · 2025-11-22

I have read and agree with the venue's withdrawal policy on behalf of myself and my co-authors.